# Spray-Dried Formulation of Epicertin, a Recombinant Cholera Toxin B Subunit Variant That Induces Mucosal Healing

**DOI:** 10.3390/pharmaceutics13040576

**Published:** 2021-04-18

**Authors:** Micaela A. Reeves, Joshua M. Royal, David A. Morris, Jessica M. Jurkiewicz, Nobuyuki Matoba, Krystal T. Hamorsky

**Affiliations:** 1Department of Pharmacology and Toxicology, University of Louisville School of Medicine, Louisville, KY 40202, USA; micaela.reeves@louisville.edu (M.A.R.); royalj@KentuckyBioProcessing.com (J.M.R.); 2James Graham Brown Cancer Center, University of Louisville School of Medicine, Louisville, KY 40202, USA; dmorris@greenlightbio.com (D.A.M.); jmmcmu02@gmail.com (J.M.J.); 3Center for Predictive Medicine, University of Louisville School of Medicine, Louisville, KY 40202, USA; 4Department of Medicine, University of Louisville School of Medicine, Louisville, KY 40202, USA

**Keywords:** cholera toxin B subunit, epicertin, spray-drying, pharmaceutical formulation, biopharmaceuticals, ulcerative colitis

## Abstract

Epicertin (EPT) is a recombinant variant of the cholera toxin B subunit, modified with a C-terminal KDEL endoplasmic reticulum retention motif. EPT has therapeutic potential for ulcerative colitis treatment. Previously, orally administered EPT demonstrated colon epithelial repair activity in dextran sodium sulfate (DSS)-induced acute and chronic colitis in mice. However, the oral dosing requires cumbersome pretreatment with sodium bicarbonate to conserve the acid-labile drug substance while transit through the stomach, hampering its facile application in chronic disease treatment. Here, we developed a solid oral formulation of EPT that circumvents degradation in gastric acid. EPT was spray-dried and packed into enteric-coated capsules to allow for pH-dependent release in the colon. A GM1-capture KDEL-detection ELISA and size-exclusion HPLC indicated that EPT powder maintains activity and structural stability for up to 9 months. Capsule disintegration tests showed that EPT remained encapsulated at pH 1 but was released over 180 min at pH 6.8, the approximate pH of the proximal colon. An acute DSS colitis study confirmed the therapeutic efficacy of encapsulated EPT in C57BL/6 mice upon oral administration without gastric acid neutralization pretreatment compared to vehicle-treated mice (*p* < 0.05). These results provide a foundation for an enteric-coated oral formulation of spray-dried EPT.

## 1. Introduction

Ulcerative colitis (UC) is a major form of inflammatory bowel disease (IBD), characterized by chronic and relapsing inflammation of the innermost layer of the colon and rectal mucosa [1,2]. Its etiology remains poorly understood and the onset is associated with a complicated interplay of genetic and environmental factors as well as gut microbiota [3]. The disease often manifests as symptoms including: bloody diarrhea, rectal bleeding, fatigue, and weight loss [4]. However, symptom presentation often varies among patients and may change over time with increasing severity of disease [5]. The Crohn’s and Colitis Foundation estimates that approximately 1.6 million Americans suffer from IBD with a total US annual financial burden between USD 14.6 billion and 31.6 billion. UC accounts for 907,000 of these cases with an annual incidence of 12.2 per 100,000 people [5]. Current Food and Drug Administration approved UC drugs aim to treat existing symptoms, maintain remission, and improve quality of life. Despite multiple treatment options available to UC patients, none of them can cure the disease, and up to a third of those with 30+ years of the disease will require surgical removal of the colon and rectum [5].

There are several classes of drugs used to treat UC [5]. Typically, UC treatment follows a step-up approach in which drug class utilization is dependent upon disease severity and response to prior therapies. The final step of this treatment strategy is surgical intervention [6,7]. Treatment with 5-aminosalicylates (5-ASAs) has long been the mainstay first-line therapy for mild-to-moderate UC [5,8]. This inflammation-blunting class of therapeutics are preferred for early-stage UC because of their generally innocuous side effect profiles, although moderate UC is often unresponsive to these agents [7]. The mild side effect profile is counteracted by the ability of patients to develop tolerance during remission maintenance and require new treatment strategies. Failure to achieve or maintain remission with 5-ASAs is typically followed by treatment with corticosteroids and steroid-sparing immunomodulators. An estimated two-thirds of patients receiving short-term steroid treatment for moderate to severe UC achieve remission. However, the risk of serious adverse effects limit long-term use of these agents [6,7,8,9]. Biologics (e.g., anti-TNFα and anti-integrin monoclonal antibodies) have traditionally been the final agents utilized to treat severe UC prior to surgical resection of the colon and rectum, although recent literature suggests the benefit of using biologics in earlier stages [10]. These drugs are effective in remission induction and maintenance in patients following previous treatment failures but are partnered with the serious side effects including severe infection and increased cancer risk [8,9]. Of particular note, fewer than half of patients treated with biologics are able to achieve mucosal healing, an endoscopic marker found to be highly predictive of sustained clinical remission, better quality of life, and decreased risk for colitis-associated colorectal cancer [11,12]. Further, biologics are typically more expensive than other therapeutic agents [13]. Therefore, there is a current unmet need in UC therapy: agents that can directly restore the damaged epithelial barrier and facilitate mucosal healing without suppressing immune function.

Mucosal healing is a major treatment goal for UC patients [11,12,14,15]. It is a complex and dynamic process involving multiple cell types including epithelial, stromal and immune cells [14]. Epithelial repair plays a crucial role in mucosal healing by rebuilding the intestinal barrier to inhibit inflammation caused by entry of bacteria into the mucosa. Since inflammation in UC is limited to the innermost layer of the colon and rectal mucosa, epithelial repair may be key to achieving mucosal healing in this subset of IBD patients [5,11]. Thus, an epithelial repair agent may fill a current treatment gap for UC. We previously found that oral administration of a plant-made recombinant variant of cholera toxin B subunit (CTB) facilitates epithelial repair and mucosal healing in dextran sulfate sodium (DSS)-induced acute and chronic colitis mouse models [15,16]. CTB is the nontoxic homopentameric component of the cholera toxin with high binding affinity to GM1 ganglioside on epithelial cells [17]. This variant, henceforth designated Epicertin (EPT), has a major modification from the parent molecule; the C-terminal hexapeptide extension containing a KDEL endoplasmic reticulum (ER) retention motif [18,19]. While the alteration did not affect the GM1-binding affinity, molecular stability or oral immunogenicity of the original molecule [18], EPT, but not wild-type CTB, induced mucosal healing in the DSS colitis model. This unique new activity, which stems from EPT’s capacity to interact with the KDEL receptor and subsequently activate the inositol-requiring enzyme 1/X-box binding protein 1 arm of an unfolded protein response in colon epithelial cells [20], lends support for the development of EPT as a new class of oral therapeutics for UC.

EPT may be administered to the colon topically or by oral gavage to alleviate DSS-induced colitis in mice [20]. Although oral medications are generally preferred by patients and increase patient adherence to treatment regimens [3,21,22], oral administration of EPT solution requires neutralization of stomach acid to prevent degradation of the protein. This is similar to the World Health Organization prequalified oral cholera vaccine, Dukoral™, which is administered in a solution following stomach neutralization with a sodium bicarbonate solution. Considering potential long-term treatment necessary for the management of UC [21], this neutralization step could be disadvantageous as it would likely lower patient adherence and ease of administration. To address this limitation, we describe here the development of a prototype enteric-coated oral formulation of EPT that allows for pH-dependent release of the drug substance in the colon, wherein the protein was spray dried and encapsulated in a gelatin capsule coated with an anionic polymer. Our results provide a foundation for further development of a novel oral biologic to facilitate colon mucosal healing in UC.

## 2. Materials and Methods 

Animals. Eight-week-old C57BL/6J, female mice were obtained from Jackson Laboratories (Bar Harbor, ME). The University of Louisville’s Institutional Animal Care and Use Committee approved all animal studies conducted in this manuscript (IACUC protocol 16713). 

EPT production. EPT was produced in *Nicotiana benthamiana* using a transient overexpression system and purified to >95% homogeneity with an endotoxin level of <3 endotoxin units per mg as described previously [19,22,23]. EPT was ultrafiltrated/diafiltrated into various buffers (Table 1) using 30,000 MWCO centrifugal devices. Phosphate buffered saline (PBS) with 100 mM mannitol was found to be the optimal buffer for spray drying of EPT. EPT at 1 mg/mL in PBS with 100 mM mannitol excipient (pH 7.2) was dehydrated using a Büchi B-290 mini spray drier with an inlet temperature of 125 °C and an outlet temperature maintained between 65 and 67 °C. The Q-Flow was 35 mm, aspirator was 90% and pump 20%. EPT powder was stored in conical tubes wrapped in parafilm under desiccation at room temperature (20–25 °C) until use. Standard EPT used for the GM1/KDEL ELISA and SEC-HPLC was produced in *Nicotiana benthamiana* using a transient overexpression system and purified to >95% homogeneity with an endotoxin level of <3 endotoxin units per mg as described previously [19,22,23].

EPT powder characterization. To measure residual moisture, a 5.9 mg portion of dried powder was incubated at 70 °C for 16 h, and the weight difference before and after heating was used to calculate moisture content. To determine solubility, dried powder was weighed and a calculated volume of milli Q water was added to reconstitute EPT powder to 1 mg/mL. The concentration of the reconstitute solution was measured by Nanodrop (Thermo Fisher Scientific) using an extinction coefficient at *A*_280_ of 0.7857. The percent solubility was calculated based on the difference in 1 mg/mL vs. the determined nanodrop concentration. 

Percent monomer was determined by size-exclusion high performance liquid chromatography (SEC-HPLC). SEC-HPLC was run as previously described [17]. Briefly, reconstituted EPT at 1 mg/mL was applied to a Tosoh TSKgel SuperSW3000 column using 100 mM sodium phosphate, pH 7.2, 150 mM sodium chloride running buffer. EPT standard (a bulk solution prepared in PBS before spray drying) was used as a control. 

GM1-capture KDEL-detection (GM1/KDEL)-ELISA. The assay was done as described in Morris et al. [24]. Plates were coated with 100 µL per well of 2 µg/mL GM1 ganglioside (Sigma Aldrich; St. Louis, MO, USA) diluted in a coating solution consisting of 3 mM sodium azide, 15 mM sodium carbonate, 35 mM sodium bicarbonate, pH 9.6. After overnight incubation (16 to 18 h) at 4 °C, plates were washed three times with PBST (0.05% Tween 20 in 1X PBS) and blocked with a blocking solution (5% non-fat dry milk, 0.05% Tween 20 in 1X PBS) for 1 h at room temperature, then washed with PBST thrice. Three-fold serially diluted, duplicate samples (100 µL/well) were added to plates in 1% PBSTM (1% dry milk, 0.05% Tween 20 in 1X PBS). Samples were incubated on plates for 1 h at 37 °C. Plates were washed and mouse anti-KDEL monoclonal antibody (Enzo Life Sciences; Farmingdale, NY, USA) diluted 1:1000 in 1% PBSTM (100 µL/well) was added; plates were then incubated at 37 °C for 1 h. Plates were washed and goat anti-mouse IgG-HRP (Southern Biotech; Birmingham, AL, USA) diluted 1:5000 in 1% PBSTM was added, followed by incubation at 37 °C for 1 h. Plates were washed a final time and developed with 3,3′,5,5′-tetramethylbenzidine substrate (TMB). The reaction was stopped with 2 N sulfuric acid and the absorbance at 450 nm was immediately measured with a BioTek plate reader.

EPT capsule package and preparation. Torpac size M gelatin capsules were packed with 2.3–2.8 mg dried EPT product per capsule (corresponding to 5 ± 1 µg of EPT) using the ProFunnel capsule filling system (Torpac; Fairfield, NJ, USA). Capsule cap and body joints were painted with 4% Eudragit L100 anionic polymer coating solution and allowed to dry for 20 min. Capsules were loaded into a size M capsule holder and dipped just past the cap and body joints into 4% Eudragit L100 coating solution (recipe recommended by Torpac) and allowed to dry for 25 min according to the capsule manufacturer instructions. Capsules were flipped and reloaded into the holder and dipped past the cap and body joints into 4% Eudragit L100 coating solution. Dipping of capsules was repeated with 20% Eudragit L100 coating solution. Eudragit L100 was the chosen polymer due to its degradation at pH 6.8, the approximate pH of the proximal colon.

Capsule disintegration test. Individual EPT-containing enteric-coated capsules (*n* = 5) were submerged in 1 N hydrochloric acid for 2 h at room temperature. Acid submerging each capsule was removed and stored separately. Capsules were washed briefly with sodium phosphate buffer (pH 6.0) followed by submersion in sodium phosphate buffer (pH 6.8). Aliquots were removed from each tube following light vortexing at 5, 10, 15, 30, 60, 90, 120, and 180 min and stored individually. EPT release from individual capsules was analyzed by CTB sandwich ELISA.

CTB sandwich ELISA. The concentration of EPT standard was measured by Nanodrop (Thermo Fisher Scientific, Waltham, MA USA). Plates were coated with 100 µL per well of 2.5 µg/mL of the rat anti-CTB monoclonal antibody 7A12B3 diluted in PBS. After overnight incubation (16 to 24 h) at 4 °C, plates were washed three times with PBST (0.05% Tween 20 in 1X PBS) and blocked with a blocking solution (3% bovine serum albumin, 0.05% Tween 20 in 1xPBS) for 2 h at room temperature, then washed with PBST in triplicate. Three-fold serially diluted, duplicate standard samples (100 µL/well) were added to plates in dilution buffer (1% bovine serum albumin, 0.05% Tween 20 in 1X PBS). Unknown samples were diluted 1:10 in dilution buffer and added to plates in duplicate (100 µL/well). Samples were incubated on plates for 1 h at room temperature. Plates were washed and rabbit anti-CTB polyclonal antibody (Abcam; Cambridge, UK) diluted 1:20,000 in dilution buffer (100 µL/well) was added; plates were then incubated at room temperature for 1 h. Plates were washed and goat anti-rabbit IgG-HRP (Southern Biotech; Birmingham, AL, USA) diluted 1:100,000 in dilution buffer was added, followed by incubation at room temperature for 1 h. Plates were washed a final time and developed with TMB. The reaction was stopped with 2 N sulfuric acid and the absorbance at 450 nm was immediately measured with a BioTek plate reader. Percent release was determined by extrapolation of calculated EPT concentrations using CTB sandwich ELISA compared to a known fixed mass of 5 µg EPT per capsule.

Acute DSS colitis model and EPT treatment. Groups of 10 female C57BL/6 mice, randomly assigned, were used. 3% (*w*/*v*) DSS (M.W. 36,000–50,000; M.P. Biomedicals, Santa Ana, CA, USA) was administered in drinking water ad libitum for 7 days. Body weights were monitored daily from the start of DSS exposure to sacrifice on day 14. On the last day of DSS exposure, animals were orally gavaged 100 µL PBS, 100 µL of EPT powder dissolved in PBS (0.03 mg/mL solution) after administration of sodium bicarbonate (200 µL of 30 mg/mL solution) as described previously, or enteric coated capsules filled with 5 µg EPT (described above) [15]. Animals recovered with normal drinking water for 7 days. Disease activity index (DAI) scores, consisting of body weight loss, fecal consistency and occult blood tests, were recorded following sacrifice and performed as previously described [25]. Distal colon tissues were fixed in neutral buffered formalin and stained with hematoxalin and eosin (H&E). Histopathological scores, a combination score comprised of crypt architecture, inflammatory infiltrate, muscle thickening and goblet cell presences scores, were determined as previously described [24,25]. Each category was ranked on a scale from 0 to 3 and summed to obtain a single histopathological damage score for each tissue.

Statistics. For all data, outliers were determined by statistical analysis using the Grubb’s test and excluded from further analysis if *p* < 0.05. Graphs were prepared and analyzed using Graphpad Prism version 5.0 (Graphpad Software, La Jolla, CA, USA). To compare two data sets, an unpaired, two-tailed Student’s *t* test was used. To compare three or more data sets, one-way ANOVA with Bonferroni’s multiple-comparison post-test. 

## 3. Results

### 3.1. Pre-Formulation Analysis

Buffers and excipients were screened to determine a combination producing optimal EPT powder. This was determined by assessing pentamer disassembly into monomer, residual moisture, yield, and solubility of powder in water, with a target product profile (TPP) of < 5% monomer, < 10% moisture, and 100 ± 10% solubility (Table 1). Of the buffers tested, PBS and PBS + 100 mM mannitol were chosen from the screened buffers based on the TPP values set for the aforementioned parameters. To assess stability of the chosen buffers, pentamer degradation and water-solubility of EPT formulated in PBS or PBS + 100 mM mannitol were analyzed over a period of three weeks (Table 2). The screened and finalized drying conditions are summarized in Table 3. Given the importance of pentamer stability to the epithelial repair activity of EPT [20], PBS + 100 mM mannitol was used to optimize drying conditions and subsequent experiments. A variety of inlet and outlet temperature range combinations were assessed to determine which would result in optimal pentamer stability. A lower inlet temperature range (116–122 °C) and higher outlet temperature range (64–67 °C) were found to result in the lowest degree of pentamer degradation. Therefore, these conditions were utilized for subsequent batch productions. 

### 3.2. Stability and Disintegration Testing of EPT Capsules

To demonstrate the stability of the chosen prototype EPT powder immediately after spray-drying we performed a GM1/KDEL-ELISA and SEC-HPLC to detect the presence of intact KDEL sequence and the conformational state of CTB pentamer, which are crucial for the mucosal healing activity of EPT [20,22]. This immunoassay and analytical analysis were repeated at 9 months post-spray dry with EPT powder stored under desiccation at 23 °C in parafilm-wrapped tubes. GM1/KDEL-ELISA results demonstrate that GM1 binding affinity of spray dried EPT was unchanged after drying (Figure 1A) and 9 months post-drying when stored under desiccation at 23 °C (Figure 1B). SEC-HPLC chromatograms illustrate the stability of EPT pentamers in 9 months post-drying (Figure 1). The small peak of 8.2% at a retention time of ~18.5 min correlates to EPT monomers (see Appendix A), indicating only slight pentamer degradation after 9 months post-drying stored under desiccation at 23 °C (Figure 1D). 

Following packing of EPT powder into gelatin capsules and coating with an enteric coating solution, pH-dependent release of EPT was assessed by a capsule disintegration assay. Capsules were coated with Eudragit S100 coating solution to prevent the release of EPT prior to arrival at the proximal colon where the intraluminal pH has increased to ~pH 6.8 after transiting through the stomach and small intestine [26,27]. The capsule disintegration assay simulated passage of the capsule through gastric acid and allowed us to analyze the release profile of EPT following exposure to pH 6.8. Recovery of EPT from capsules was determined by a CTB detection sandwich ELISA, which has the capacity to detect both GM1-binding pentamer and disassembled CTB molecular species unlike GM1-capture ELISA (Appendix A), as EPT dissociation could occur if the capsule content was prematurely discharged and exposed to low pH conditions (Appendix A). The data revealed that EPT was released from the polymer-coated capsules in a pH-dependent manner (Figure 2). No EPT was detectable at pH 1.0, suggesting that the enteric coating prevented the release of EPT from capsules prior to expected release at pH 6.8. The same coating solution and method was used to prepare capsules used in the following acute colitis model.

### 3.3. Efficacy of EPT Enteric-Coated Capsules in an Acute Colitis Model

The efficacy of EPT delivered by this oral capsule formulation was assessed compared to standard EPT in an acute DSS colitis mouse model [15,20]. In this model, animals were exposed to 3% DSS ad libitum in drinking water for 7 days at which time mice were dosed 5 µg resuspended dried EPT (EPT powder solution) or PBS via oral gavage after administration of sodium bicarbonate. A third group of mice were dosed with an enteric-coated capsule containing 5 µg EPT with no administration of sodium bicarbonate. All animals were monitored an additional 7 days following DSS cessation. Compared to PBS, DAI scores were decreased in mice administered a EPT capsule directly as well as sodium bicarbonate followed by reconstituted EPT powder (*p* < 0.05 and *p* < 0.001, respectively) (Figure 3). To corroborate the aforementioned results, we performed a histopathological evaluation to assess the presence of hallmark colitis markers, such as alterations in crypt height and loss, epithelial barrier disruption, and immune cell infiltration, in hematoxalin and eosin (H&E) stained tissues. EPT powder solution administration following gastric acid neutralization and encapsulated EPT similarly protected mice from DSS-induced acute colitis. Treatment with EPT by capsule administration or oral gavage following gastric acid neutralization protected mice from crypt loss and distortion, inflammatory cell infiltrates, muscle thickening, and goblet cell loss (Figure 4A, B). However, few crypt structural alterations, primarily crypt branching, were noted in tissues from both EPT treatment groups. Taken together, these results support equivalence between EPT treatment in solution following acid neutralization and encapsulated EPT treatment.

## 4. Discussion

EPT is a variant of the nontoxic component of the cholera toxin that exhibits unique mucosal healing activity in the colon [16,18,21]. Previous studies examining the therapeutic potential of EPT in mouse colitis models have primarily focused on one route of administration: oral gavage. An issue with this route of administration, however, is the need to neutralize gastric acid with sodium bicarbonate prior to gavage as CTB is acid labile. This is a drawback when moving forward with development of EPT as a therapeutic for UC and determining a final drug product formulation. Although oral agents are typically preferred by patients undergoing treatment for chronic diseases such as UC, gastric acid neutralization requirements could potentially affect patient outcomes by lowering adherence and ease of administration. Therefore, we aimed to develop a prototype solid oral formulation that would allow EPT to circumvent gastric acid degradation and allow for topical administration to affected areas. We described herein an encapsulated spray-dried drug substance coated with an enteric coating to allow for pH-dependent release of EPT at the colon.

A major technical advance made in the present study towards a solid oral formulation is the establishment of the method of drying the drug substance. Drying of pharmaceuticals is a long-implemented practice commonly used to enhance final drug product for a variety of purposes; examples of the benefits of biopharmaceutical dehydration include: handling and storage improvement, decrease in transportation cost, improved stability and aid in development of modified or delayed release particles [28,29]. It is known that proteins are more stable in solid rather than liquid form [30,31,32,33]. Use of solid formulations can greatly increase shelf-life and reduce storage regulations, saving patients and manufacturers money in lost production costs due to expired product. Further, oral capsules filled with dried protein may be coated with a time- or pH-dependent coating to allow for targeted release in the GI tract [29]. This is especially useful when administering CTB orally as it allows for release at the affected site. Without this coating, orally administered pentameric CTB would degrade into nonfunctional monomers upon exposure to the stomach acid. Dehydrating CTB is one solution to this issue.

Previously, CTB has been dehydrated by a variety of methods. A freeze-dried inactivated whole-cell oral cholera vaccine was formulated in attempts to optimize delivery of mass quantities of vaccine to low-income countries [34]. This formulation elicited strong serum and gut mucosal anti-LPS antibody responses in immunized mice; these responses were comparable to those achieved with equivalent liquid formulation [34]. The dry formulation is beneficial in substantially reducing package volumes and weights when delivering product to areas in need of mass vaccination. Further, CTB has been successfully spray-dried in the form of heat-killed *Vibrio cholerae*-containing microparticles [35]. We utilized the benefits of the spray-drying process with EPT to develop a more optimal oral formulation of the protein. When a protein is spray-dried, conditions need to be tailored to the protein being dried since materials undergo some thermal stress which can result in protein degradation; hence, identification of ideal heating conditions is critical. CTB pentamer degradation occurs approximately between 66 and 78 °C [36,37,38]. Therefore, we screened outlet temperature ranges between 61 and 67 °C to maintain stability of functional EPT pentamer. Since we achieved optimal TPP parameters with outlet temperatures closer to 67 °C, testing outlet temperatures closer to 78 °C could possibly result in a further improved powder by solubility or moisture content. The most relevant source of stress during spray-drying results from the dehydration process, therefore the addition of excipients to the liquid solution prior to spray drying is crucial to replace the hydrogen bonding that exists in an aqueous environment [29]. In this study, a screen was developed to identify optimal excipient conditions to improve target profile parameters.

A buffer and excipient screen to produce an ideal dried EPT powder based on a set TPP (<5% monomer, <10% moisture, and 100 ± 10% solubility). Addition of a common excipient (mannitol) decreased the presence of EPT monomer from approximately 5% to 3% (Table 2 and Table 3). It is unsurprising that addition of mannitol improved stability of EPT pentamer, as it is often used in dried pharmaceuticals due to its thermostability [39,40,41]. PBS + 100 mM mannitol outperformed all other screened buffers in the aforementioned TPP categories and was therefore chosen as the formulation buffer. All TPP categories were met, however, loss of drug product was a consistent problem as 70% was the maximum recovery among all buffers tested (data not shown). Other studies using this particular spray drier consistently report yields below 50% [42]. Although one issue with this instrument is in aspects of the design, another manageable issue is identifying an ideal inlet to outlet temperature ratio for the protein of interest [42]. This limitation allows room for improvement moving forward in formulation development. Stability of EPT powder produced in the chosen buffer was confirmed by measuring monomer content and percent solubility each week for a total of three weeks (Table 2 and Table 3). Low hygroscopicity of mannitol likely had a positive impact on moisture content; it is known that mannitol is an ideal excipient to minimize moisture in a dried formulation [40,41]. Monomer content and solubility remained stable around 3% and 98%, respectively, over the course of three weeks. Mannitol seems to have a protective effect on EPT pentamers, possibly due its thermostability. This study, however, lacked investigation of other thermoprotective agents such as dextrose, trehalose and lactose as a potential excipient. Lactose is the most commonly utilized excipient in spray drying [39,43]. Mannitol and lactose are both attractive excipients as they are soluble in water and are non-toxic. Lactose has an advantage to mannitol as it is more economical, although it should be noted that lactose has a higher hygroscopicity which can hinder stability of the dried product [39,41]. Further, lactose is more likely to react with active pharmaceutical ingredients as it is a reducing sugar, whereas mannitol exhibits a strong inertness [39]. 

Upon selection of buffer composition, spray dry parameters were further investigated. Spray dry parameters were optimized by screening a combination of inlet and outlet temperatures to determine a combination producing EPT powder with the best possible TPP values (Table 3). It has been suggested that a high inlet temperature to outlet temperature ratio might be the key to maximizing yield. Our results are consistent with this claim as the highest inlet:outlet temperature condition tested achieved optimal TPP values (Table 3). We also demonstrated via SEC-HPLC and GM1-KDEL detection ELISA that spray dried EPT maintains GM1 binding affinity and remains stable under dry conditions for up to 9 months (Figure 1). This combination of factors indicates that EPT powder encapsulated immediately post-dry and after 9 months should exhibit similar effects upon administration.

The oral formulation presented here was designed for direct delivery of EPT to the target site by utilizing an anionic polymer coating that makes pH-dependent release at the colon possible. The in vitro disintegration test suggested that not all capsules released 100% of encased EPT (Figure 2). It is possible that there are inconsistencies in the thickness of the enteric coating around the capsules, leading to variations in release profiles. The combination of dipping capsules by and viscosity of the solution could lead to disparities between capsule coatings. This could be ameliorated by coating the capsules using an automated system that would likely be utilized when coating capsules at an industrial scale. Further, it was observed during disintegration testing that capsules tend to fold in on themselves when free-floating in solution. This is an unlikely issue in vivo as movement through the GI tract would prevent the folding over of capsules. In support of this hypothesis, the acute DSS study showed that the enteric-coated EPT capsule ameliorated acute DSS-induced colitis, indicating that the protein was successfully released from the capsules at the site of mucosal damage (Figure 3 and Figure 4). The histopathological results described here are also consistent with previous findings in acute and chronic DSS colitis studies evaluating EPT treatment [16,17,21]. Taken together, these results support further development of the capsule formulation described herein as this treatment does not require gastric acid neutralization, which would theoretically ease difficulty of administration and boost patient adherence in UC patients. 

## Figures and Tables

**Figure 1 pharmaceutics-13-00576-f001:**
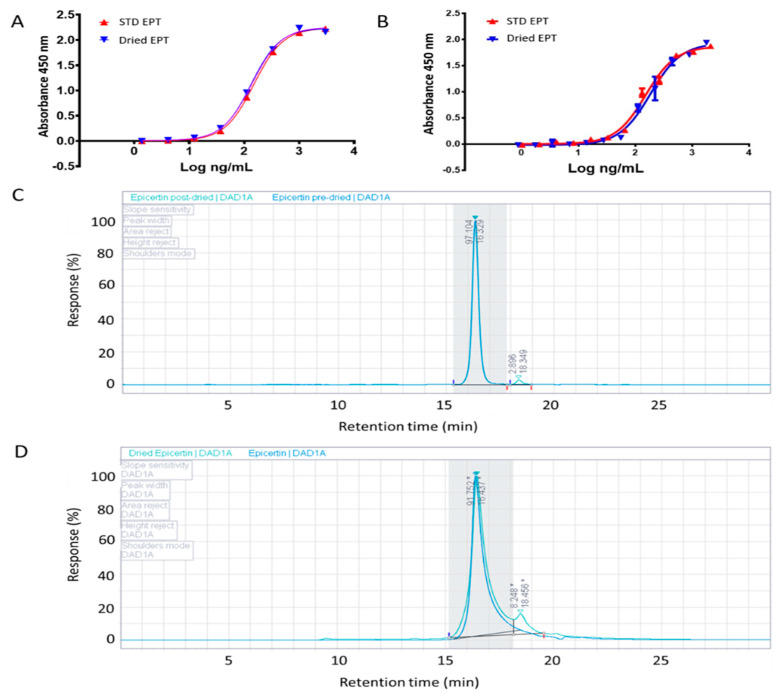
Stability assessment of spray dried EPT. The production of EPT standard is described in the methods section. The stability of dried EPT post-drying and after 9 months was assessed by GM1−capture KDEL−detection ELISA and SEC-HPLC. A representative binding curve of spray dried EPT is shown for (**A**) one day post-drying and (**B**) after 9 months stored at 23 °C in a desiccator, compared to an EPT standard. SEC-HPLC chromatogram of (**C**) non-dried EPT (blue) and EPT one day post-drying (green) and (**D**) dried EPT (green) after 9 months stored at 23 °C in a desiccator (EPT standard is in blue). After 9 months dried EPT contained 91.8% pentamer and 8.2% monomer (* represents a line drop as the two peaks are not completely resolved). The resolution value of the two peaks is 1.6 (determined by OpenLab CDS 2.1 software, Agilent Technologies, Santa Clara, CA, USA).

**Figure 2 pharmaceutics-13-00576-f002:**
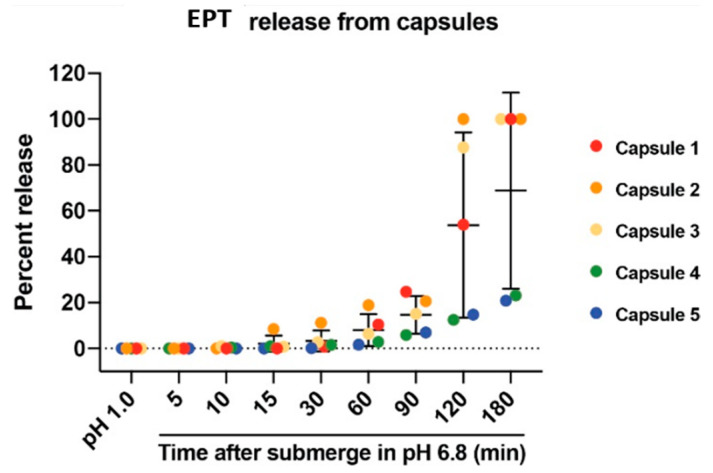
EPT released from enteric-coated capsules is pH-dependent. Release of EPT was measured by CTB sandwich ELISA. EPT releases from capsules only after submersion at pH 6.8. Percent of EPT release from capsules after 2 h submersion in 0.1 N HCl, and at t = 5, 10, 15, 30, 60, 90, 120, and 180 min after pH shift to 6.8. *n*= 5. Percent release was determined by extrapolation of calculated EPT concentrations using CTB sandwich ELISA compared to a known fixed mass of 5 µg EPT per capsule.

**Figure 3 pharmaceutics-13-00576-f003:**
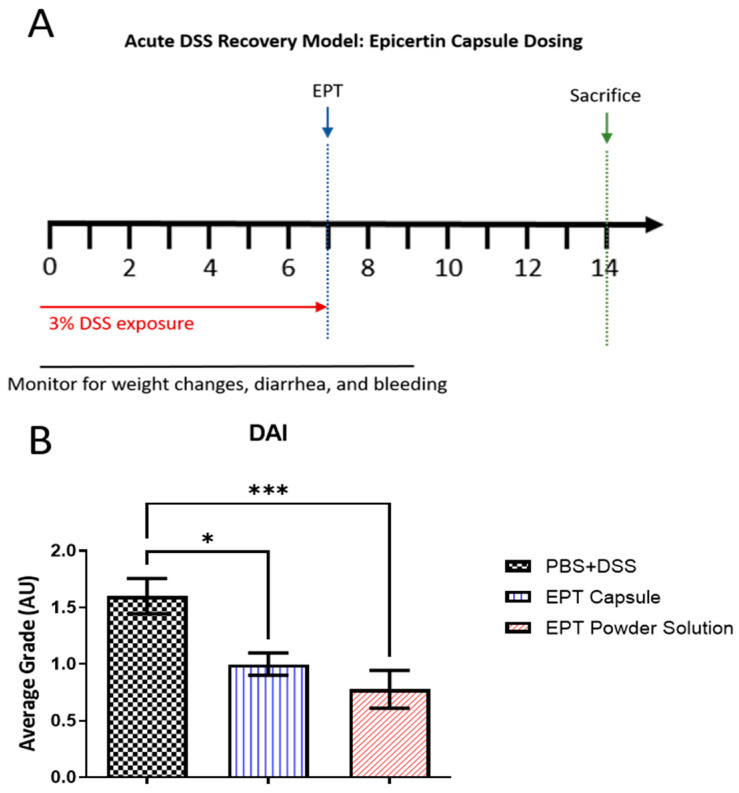
Enteric-coated EPT capsules mitigate acute DSS colitis in mice. (**A**) Study design. (**B**) Mice were dosed with an EPT capsule (*n* = 10), pre-dissolved EPT powder following gastric acid neutralization (EPT powder solution; *n* = 9), or capsule vehicle control (*n* = 10) on day 7 following DSS exposure. DAI scores were determined on day 14 as a combined measure of body weight recovery, stool consistency, and blood in stool; data are shown as mean ± SEM. ** p* < 0.05, *** *p* < 0.001, one-way ANOVA with Bonferroni’s multiple comparisons test.

**Figure 4 pharmaceutics-13-00576-f004:**
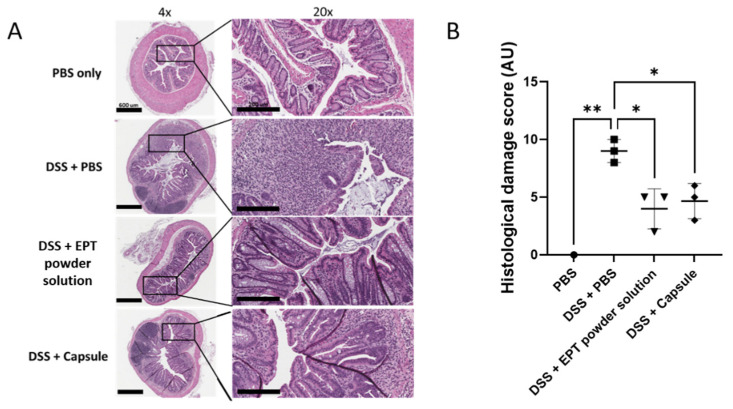
Treatment with encapsulated EPT mitigates acute colitis in mice. Encapsulated EPT (DSS + Capsule) treatment protected mice from histological damage similarly to treatment with EPT powder solution following gastric acid neutralization (DSS + EPT powder solution). (**A**) Representative 4× (left) and 20× (right) photomicrographs of H&E-stained distal colon tissues from each treatment group. (**B**) Histological damage scores of each treatment group in the DSS acute colitis study. * *p* < 0.05, ** *p* < 0.01; one-way repeated measures ANOVA with Bonferroni’s multiple comparisons test.

**Table 1 pharmaceutics-13-00576-t001:** Determination of optimal buffer excipient for EPT powder production.

Buffer	% Monomer	% Moisture	% Solubility
PBS	4.9	2.9	106
PBS, 20 mM Mannitol	4.5	10.3	94
PBS, 100 mM Mannitol	3.1	0	97
PBS, 150 mM Mannitol	9.1	7.5	99
PBS, 250 mM Mannitol	6.4	4.1	87
30 mM Phosphate, pH 7	3.2	31.5	67
30 mM Phosphate, 20 mM Mannitol, pH 7	1.1	14.7	81
30 mM Phosphate, 100 mM Mannitol, pH 7	5.8	2.3	97
30 mM Phosphate, 150 mM Mannitol, pH 7	5.8	3.9	96
30 mM Phosphate, 250 mM Mannitol, pH 7	6.7	5.4	91
30 mM Phosphate, pH 7.5	6.1	20	79
30 mM Phosphate, 20 mM Mannitol, pH 7.5	2.4	21.3	82
30 mM Phosphate, 100 mM Mannitol, pH 7.5	5.2	1	88
30 mM Phosphate, 150 mM Mannitol, pH 7.5	7.0	2.1	92
30 mM Phosphate, 250 mM Mannitol, pH 7.5	7.3	3.1	64
88 mM Phosphate, 20 mM Mannitol, pH 7	2.1	18.4	96
TARGET	<5%	<10%	100 ± 10

**Table 2 pharmaceutics-13-00576-t002:** Stability of EPT powder in chosen buffer excipients.

Buffer	% Monomer	% Solubility
PBS	4.9	106
PBS, 100 mM Mannitol	3.1	97
**1 Week**		
PBS	5.9	110
PBS, 100 mM Mannitol	4.0	97
**2 Weeks**		
PBS	4.8	110
PBS, 100 mM Mannitol	3.3	99
**3 Weeks**		
PBS	5.9	110
PBS, 100 mM Mannitol	3.4	99

**Table 3 pharmaceutics-13-00576-t003:** Optimization of spray-dry parameters.

Buffer	% Monomer	% Moisture	% Solubility	Q-Flow (min)	Inlet (°C)	Outlet (°C)	Aspirator (%)	Pump (%)
PBS	12.3	9.7	97	-	121	66–67	90	20
PBS	8.0	3.7	99	-	120	61–64	90	20
PBS	4.9	2.9	106	35	118–125	61–65	90	20
PBS	11	-	99	34	121–123	61–63	90	20
PBS, 100 mM Mannitol	3.1	0	97	35	120–123	62–64	90	20
PBS, 100 mM Mannitol	9	-	119	34	121–124	63–66	90	20
PBS, 100 mM Mannitol *	1.0	9.5	97	35	116–122	64–67	90	20
PBS, 100 mM Mannitol @ 5 mg/mL *	1.1	4.9	96	35	118–122	61–63	90	20

* Finalized drying conditions.

## Data Availability

The data presented in this study are available within the article and its Appendix A.

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
