# Peer review of "Spray-Dried Formulation of Epicertin, a Recombinant Cholera Toxin B Subunit Variant That Induces Mucosal Healing"

_pharmaceutics, 2021, doi:10.3390/pharmaceutics13040576_

Round 1

Reviewer 1 Report

Comments

  1. In the title “Spray dry formulation of EPICERTIN, a recombinant cholera 2 toxin B subunit variant that induces mucosal healing”. Write Spray dried in place of spray dry.
  2. Why in vitro release studies were not performed to check the sustained release ?.
  3. Authors report mucosal healing by Spray dried formulation of EPICERTIN, Is it local healing or systemic healing?
  4. Authors are advised to add reference “Anwer MK, Ahmed MM, Aldawsari MF, Alshahrani S, Fatima F, Ansari MN, Rehman NU, Al-Shdefat RI. Eluxadoline Loaded Solid Lipid Nanoparticles for Improved Colon Targeting in Rat Model of Ulcerative Colitis. Pharmaceuticals (Basel). 2020 Sep 19;13(9):255.” In introduction part.

Author Response

Response to Reviewer 1:

  1. In the title “Spray dry formulation of EPICERTIN, a recombinant cholera 2 toxin B subunit variant that induces mucosal healing”. Write Spray dried in place of spray dry.
  • The title has been updated to reflect the proposed change.

  1. Why in vitro release studies were not performed to check the sustained release ?
  • An in vitro disintegration test was performed to assess sustained release and is described in Figure 2.

  1. Authors report mucosal healing by Spray dried formulation of EPICERTIN, Is it local healing or systemic healing?
  • In this study we confirmed the mucosal healing activity of EPICERTIN in the distal colon where DSS caused epithelial damage and inflammation. Our preliminary in vivo toxicology results in healthy and colitic mice indicate little-to-no systemic absorption following intracolonic administration of Epicertin (manuscript in preparation). Therefore, although there remains a possibility for some indirect systemic impacts, we believe that EPICERTIN primarily elicits local healing to the site of administration.

  1. Authors are advised to add reference “Anwer MK, Ahmed MM, Aldawsari MF, Alshahrani S, Fatima F, Ansari MN, Rehman NU, Al-Shdefat RI. Eluxadoline Loaded Solid Lipid Nanoparticles for Improved Colon Targeting in Rat Model of Ulcerative Colitis. Pharmaceuticals (Basel). 2020 Sep 19;13(9):255.” In introduction part.

The suggested reference was included in the introduction in line 38.

Reviewer 2 Report

The manuscript is interesting and has novelty. It needs some corrections. Increase the clarity of the Supply figure 1 and also write the importance of the figure for the result outcome.

  1. As per the journal guidelines, results and discussion should be written next to the introduction, then followed materials and methods section. Format the manuscript as per the journal guidelines.
  2. In the title, EPICERTIN should be in small letters.
  3. Expand the DSS in abstract line #16. Write the significance level of therapeutic efficacy and controls used for the study.
  4. Delete CTB, IBD, mucosal healing from the key words.
  5. The introduction of the manuscript is very well written and very informative on UC.
  6. In ref 2, the year of report release is missing. Please clarify the report findings.
  7. Spell out FDA, WHO, PBS.
  8. Line# 49, change the abbreviation to 5-ASAs.
  9. Line #92 and 93. Separate the statements with ‘.’
  10. Dukoral – use trademark symbol.
  11. Write the animal approval protocol number.
  12. Separate the methods section wise. Give the sections as per the journal format.
  13. Methods - The spray-dried process parameters are missing in the methods.
  14. Eudragit L100 – write the details of the polymer. What is the basis for selecting this, pH and also colon targeting ability? Why 4% of Eudragit used, provide the reference or preliminary studies
  15. Write the units of 3% DSS.
  16. Write the pH of PBS used for EPT solution.
  17. Statistics – p should be small.
  18. one-way ANOVA with Bonferroni’s multiple-comparison post-test. What is the reason for this analysis used for comparison?
  19. What is the reason for selecting mannitol? Dextrose or trehalose also behave best for the production.
  20. Figure 1 A and B, write the units of absorbance in Y axis. In figure 1B, why the SD values added in between Log 2-3 ng/mL values. C and D difficult to read. Improve the quality of the figures. In D figure, there is peak tailing observed and also overlap with main peak before returns to baseline. What is the peak resolution value after 9 months of the stability sample?
  21. Line # 278, ad libitum – italics.
  22. In methods, third group treated with 3 ug of EPT loaded enteric coated capsule. But results mentioned, 5 ug of EPT loaded enteric coated capsules.
  23. Write the significance level in line #283-284.

Author Response

Response to Reviewer 2:

The manuscript is interesting and has novelty. It needs some corrections. Increase the clarity of the Supply figure 1 and also write the importance of the figure for the result outcome.

  • We appreciate the reviewer’s comments and have made the necessary changes to supplementary figure 1 by adding the phrase “ Evidence for EPT degradation at pH 1.0, These results demonstrate the need for a pH-dependent oral formulation of EPT. “ to the figure legend. We understand the need to write the importance of the figure, however this has already been described in lines 244-247 and lines 264-267.

  1. As per the journal guidelines, results and discussion should be written next to the introduction, then followed materials and methods section. Format the manuscript as per the journal guidelines.
  • We acknowledge the author’s concerns; however, we politely decline to make the described change as the 2021 manuscript template provided by MDPI Pharmaceutics lists the sections in the following order: introduction, materials and methods, and results. Further, other articles in this journal follow the same format.

   2. In the title, EPICERTIN should be in small letters.

  • This change has been reflected in the title.

3. Expand the DSS in abstract line #16. Write the significance level of therapeutic efficacy and controls used for the study.

  • The first “DSS” was expanded to “dextran sodium sulfate” in the abstract and significance levels and controls were also added. This change increased the abstract beyond the word limit therefore the second half of the last sentence was deleted.

4. Delete CTB, IBD, mucosal healing from the key words.

  • The keywords “CTB”,“IBD”, and “mucosal healing” were removed from the keywords.

5. The introduction of the manuscript is very well written and very informative on UC.

  • We thank the reviewer for their comment and positive feedback.

6. In ref 2, the year of report release is missing. Please clarify the report findings.

  • The year of the report (2014) was added to the reference section.

7. Spell out FDA, WHO, PBS.

  • These abbreviations were spelled out as suggested.

8. Line# 49, change the abbreviation to 5-ASAs.

  • The proposed change was adopted.

9. Line #92 and 93. Separate the statements with ‘.’

  • The proposed change was adopted.

10. Dukoral – use trademark symbol.

  • The proposed change was adopted.

11. Write the animal approval protocol number.

  • The approved IACUC protocol number “IACUC 16713” was added to the animals section of the materials and methods (line 109).

12. Separate the methods section wise. Give the sections as per the journal format.

  • The methods are separated into sections according to the journal template. However, the section titles at the beginning of each section are now in bold.

13. Methods - The spray-dried process parameters are missing in the methods.

  • The spray-dried process parameters are listed in the materials and methods section under the EPT production section (lines 116-119).

14. Eudragit L100 – write the details of the polymer. What is the basis for selecting this, pH and also colon targeting ability? Why 4% of Eudragit used, provide the reference or preliminary studies

  • The following sentence was added to the EPT capsule package and preparation section (line 164 - 165) of the materials and methods: “Eudragit L100 was the chosen polymer due to its degradation at pH 6.8, the approximate pH of the proximal colon.”
  • In the EPT capsule package and preparation section of the materials and methods the sentence “Capsules were loaded into a size M capsule holder and dipped just past the cap and body joints into 4% Eudragit L100 coating solution and allowed to dry for 25 min” was changed to “Capsules were loaded into a size M capsule holder and dipped just past the cap and body joints into 4% Eudragit L100 coating solution (recipe recommended by Torpac) and allowed to dry for 25 min according to the capsule manufacturer instructions.” (line 160-161)

15. Write the units of 3% DSS.

  • The units of 3% DSS “(w/v)” was added to the “Acute DSS colitis model and EPT treatment” section of the materials and methods (line 193).

16. Write the pH of PBS used for EPT solution.

  • The pH of PBS solution is described in line 116.

17. Statistics – p should be small.

  • The proposed change was adopted throughout the manuscript.

18. one-way ANOVA with Bonferroni’s multiple-comparison post-test. What is the reason for this analysis used for comparison?

  • The Bonferroni’s multiple-comparison post-test was used to provide a pairwise comparison of the means by comparing all pairs. It is one of the most frequently used multiple comparison tests following one-way ANOVA in biomedical research (Mishra et al. Ann Card Anaesth 2019;22(4):407-411; Lee & Lee. Korean J Anesthesiol 2018;71(5):353-360; McHigh. Biochem Med 2011;21(3):203-9) and deemed appropriate for the experimental design in the present study.

19. What is the reason for selecting mannitol? Dextrose or trehalose also behave best for the production.

  • The rationale for selecting mannitol as an excipient is described in lines 367-370 and lines 380-384. However, we respect this comment and expanded the following sentence in the manuscript “This study, however, lacked investigation of “other thermoprotective agents such as dextrose, trehalose and” lactose as a potential excipient (lines 385-386).

20. Figure 1 A and B, write the units of absorbance in Y axis. In figure 1B, why the SD values added in between Log 2-3 ng/mL values. C and D difficult to read. Improve the quality of the figures. In D figure, there is peak tailing observed and also overlap with main peak before returns to baseline. What is the peak resolution value after 9 months of the stability sample?

  • Figure 1 has been updated to reflect the suggested changes and improve the quality of the SEC-HPLC chromatograms. SD values are present on all data points for Figure 1A and 1B. The error is too small to see except between log 2-3 ng/mL values. The peak resolution was 1.6. This information was added to figure legend for 1D.

21. Line # 286, ad libitum – italics.

  • The proposed change was adopted.

22. In methods, third group treated with 3 ug of EPT loaded enteric coated capsule. But results mentioned, 5 ug of EPT loaded enteric coated capsules.

  • In line 198, “...or enteric coated capsules filled with 3 µg EPT...” was changed to “...or enteric coated capsules filled with 5 µg EPT...”. The first version describing a 3 µg capsule was a typo.

23. Write the significance level in line #292-293.

  • The significance levels were added per the reviewer’s suggestion.

Round 2

Reviewer 2 Report

The manuscript modified as per the comments and satisfactory.